# Riemannian batch normalization for SPD neural networks

**Daniel Brooks**
Thales Land and Air Systems, BU ARC
Limours, FRANCE
Sorbonne Université, CNRS, LIP6
Paris, FRANCE

**Olivier Schwander**
Sorbonne Université, CNRS, LIP6
Paris, FRANCE

**Frédéric Barbaresco**
Thales Land and Air Systems, BU ARC
Limours, FRANCE

**Jean-Yves Schneider**
Thales Land and Air Systems, BU ARC
Limours, FRANCE

**Matthieu Cord**
Sorbonne Université, CNRS, LIP6
Paris, FRANCE

## Abstract

Covariance matrices have attracted attention for machine learning applications due to their capacity to capture interesting structure in the data. The main challenge is that one needs to take into account the particular geometry of the Riemannian manifold of symmetric positive definite (SPD) matrices they belong to. In the context of deep networks, several architectures for these matrices have recently been proposed. In our article, we introduce a Riemannian batch normalization (batchnorm) algorithm, which generalizes the one used in Euclidean nets. This novel layer makes use of geometric operations on the manifold, notably the Riemannian barycenter, parallel transport and non-linear structured matrix transformations. We derive a new manifold-constrained gradient descent algorithm working in the space of SPD matrices, allowing to learn the batchnorm layer. We validate our proposed approach with experiments in three different contexts on diverse data types: a drone recognition dataset from radar observations, and on emotion and action recognition datasets from video and motion capture data. Experiments show that the Riemannian batchnorm systematically gives better classification performance compared with leading methods and a remarkable robustness to lack of data.

## 1 Introduction and related works

Covariance matrices are ubiquitous in any statistical related field but their direct usage as a representation of the data for machine learning is less common. However, it has proved its usefulness in a variety of applications: object detection in images [46], analysis of Magnetic Resonance Imaging (MRI) data [41], classification of time-series for Brain-Computer Interfaces [8] (BCI). It is particularly interesting in the case of temporal data since a global covariance matrix is a straightforward way to capture and represent the temporal fluctuations of data points of different lengths. The main difficulty is that these matrices, which are symmetric positive definite (SPD), cannot be seen as points in a Euclidean space: the set of SPD matrices is a curved Riemannian manifold, thus tools from non-Euclidean geometry must be used; see [10] for a plethora of theoretical justifications and

properties on the matter. For this reason most of classification methods (which implicitly make the hypothesis of a Euclidean input space) cannot be used successfully.

Interestingly, relatively simple machine learning techniques can produce state-of-art results as soon as the particular Riemannian geometry is taken into account. This is the case for BCI: [8, 7] use nearest barycenter (but with Riemannian barycenter) and SVM (but on the tangent space of the barycenter of the data points) to successfully classify covariances matrices computed on electroencephalography multivariate signals (EEG); in the same field,[51] propose kernel methods for metric learning on the SPD manifold . Another example is in MRI, where [41, 4] develop a $k$-nearest neighbors algorithm using a Riemannian distance. Motion recognition from motion skeletal data also benefits from Riemannian geometry, as exposed in [16], [30] and [29]. In the context of neural networks, an architecture (SPDNet) specifically adapted for these matrices has been proposed [28]. The overall aspect is similar to a classical (Euclidean) network (transformations, activations and a final stage of classification) but each layer processes a point on the SPD manifold; the final layer transforms the feature manifold to a Euclidean space for further classification. More architectures have followed, proposing alternatives to the basic building blocks: in [23] and [27], a more lightweight transformation layer is proposed; in [52] and [18], the authors propose alternate convolutional layers, respectively based on multi-channel SPD representation and Riemannian means; a recurrent model is further proposed in [19]; in [37] and [36], an approximate matrix square-root layer replaces the final Euclidean projection to lighten computational complexity. In [15], a SPD neural network is appended to a fully-convolutional net to improve on performance and robustness to data scarcity. All in all, most of the developments focus on improving or modifying existing blocks in an effort to converge to their most relevant form, both theoretically and practically; in this work, we propose a new building block for SPD neural networks, inspired by the well-known and well-used batch normalization layer [31]. This layer makes use of batch centering and biasing, operations which need to be defined on the SPD manifold. As an additional, independent SPD building block, this novel layer is agnostic to the particular way the other layers are computed, and as such can fit into any of the above architectures. Throughout the paper we choose to focus on the original architecture proposed in [28]. Although the overall structure of the original batchnorm is preserved, its generalization to SPD matrices requires geometric tools on the manifold, both for the forward and backward pass. In this study, we further assess the particular interest of batch-normalized SPD nets in the context of learning on scarce data with lightweight models: indeed, many fields are faced with costly, private or evasive data, which strongly motivates the exploration of architectures naturally resilient to such challenging situations. Medical imagery data is well-known to face these issues [41], as is the field of drone radar classification [14], which we study in this work: indeed, radar signal acquisition is prohibitively expensive, the acquired data is usually of confidential nature, and drone classification in particular is plagued with an ever-changing pool of targets, which we can never reasonably hope to encapsulate in comprehensive datasets. Furthermore, hardware integration limitations further motivate the development of lightweight models based on a powerful representation of the data. As such, our contributions are the following:

- a Riemannian batch normalization layer for SPD neural networks, respecting the manifold's geometry;

- a generalized gradient descent allowing to learn the batchnorm layer;

- extensive experimentations on three datasets from three different fields, (experiments are made reproducible with our open-source PyTorch library, released along with the article).

Our article is organized as follows: we first recall the essential required tools of manifold geometry; we then proceed to describe our proposed Riemannian batchnorm algorithm; next, we devise the projected gradient descent algorithm for learning the batchnorm; finally, we validate experimentally our proposed architecture.

## 2 Geometry on the manifold of SPD matrices

We start by recalling some useful geometric notions on the SPD manifold, noted $\mathcal{S}_*^+$ in the following.

## 2.1 Riemannian metrics on SPD matrices

In a general setting, a Riemannian distance $\delta_{\mathfrak{R}}(P_1, P_2)$ between two points $P_1$ and $P_2$ on a manifold is defined as the length of the geodesic $\gamma_{P_1 \to P_2}$, i.e. the shortest parameterized curve $\xi(t)$, linking them:

$$\delta_{\mathfrak{R}}(P_1, P_2) = \inf_{\xi \mid (\xi(0) = P_1, \xi(1) = P_2)} \int_0^1 ds(t)dt$$
$$ds(t)^2 = \dot{\xi}(t)^T \boldsymbol{F}_{\xi(t)} \dot{\xi}(t) \tag{1}$$

In the equation above, $ds$ is the infinitesimal distance between two close points and $\boldsymbol{F}$ is the metric tensor, which defines a local metric at each point on the manifold. $\dot{\xi}$ is the velocity of the curve, sometimes noted $d\xi$. For manifolds of exponential family distributions, $\boldsymbol{F}$ is none other than the Fisher information matrix (FIM) (the inverse of which defines well-known Cramer-Rao bound), which is the Hessian matrix of the entropy. This connection between entropy and differential metrics was first made in 1945 by C.R. Rao [42] and in 1943 by M. Fréchet [26], and further axiomatized in 1965 by N.N. Chentsov [17]. Then, in a 1976 confidential report cited in [5], S.T. Jensen derived the infinitesimal distance between two centered multivariate distributions $ds(\xi)^2 = \frac{1}{2} tr(\xi^{-1}\dot{\xi}\xi^{-1}\dot{\xi})$. Such distributions being defined entirely by the covariance matrix, they are isomorphic to the SPD manifold, so the integration of $ds$ along the geodesic leads to the globally-defined natural distance on $\mathcal{S}_*^+$ [38], also called affine-invariant Riemannian metric (AIRM) [41], which can be expressed using the standard Frobenius norm $\|\cdot\|_F$:

$$\delta_{\mathfrak{R}}(P_1, P_2) = \frac{1}{2}\|log(P_1^{-\frac{1}{2}} P_2 P_1^{-\frac{1}{2}})\|_F \tag{2}$$

The interested reader may note that while the above metric is the correct one from the information geometric viewpoint, it is notoriously computation-heavy. Other metrics or divergences, either closely approximate it or provide an alternate theoretical apporach, while contributing the highly desirable property of lightweight computational complexity, especially in the modern context of machine learning. Notable examples may include the usage of the Fisher-Bures metric [45], the Bregman divergence [11, 44, 6], and optimal transport [3].

Another matter of importance is the definition of the natural mappings to and from the manifold and its tangent bundle, which groups the tangent Euclidean spaces at each point in the manifold. At any given reference point $P_0 \in \mathcal{S}_*^+$, we call logarithmic mapping $Log_{P_0}$ of another point $P \in \mathcal{S}_*^+$ at $P_0$ the corresponding vector $S$ in the tangent space $\mathcal{T}_{P_0}$ at $P_0$. The inverse operation is the exponential mapping $Exp_{P_0}$. In $\mathcal{S}_*^+$, both mappings (not to be confused with the matrix $log$ and $exp$ functions) are known in closed form [2]:

$$\forall S \in \mathcal{T}_{P_0}, Exp_{P_0}(S) = P_0^{\frac{1}{2}} exp(P_0^{-\frac{1}{2}} S P_0^{-\frac{1}{2}}) P_0^{\frac{1}{2}} \in \mathcal{S}_*^+ \tag{3a}$$

$$\forall P \in \mathcal{S}_*^+, Log_{P_0}(P) = P_0^{\frac{1}{2}} log(P_0^{-\frac{1}{2}} P P_0^{-\frac{1}{2}}) P_0^{\frac{1}{2}} \in \mathcal{T}_{P_0} \tag{3b}$$

## 2.2 Riemannian barycenter

The first step of the batchnorm algorithm is the computation of batch means; it may be possible to use the arithmetic mean $\frac{1}{N}\sum_{i \leq N} P_i$ of a batch $\mathcal{B}$ of $N$ SPD matrices $\{P_i\}_{i \leq N}$, we will rather use the more geometrically appropriate Riemannian barycenter $\mathfrak{G}$, also known as the Fréchet mean [48], which we note $Bar(\{P_i\}_{i \leq N})$ or $Bar(\mathcal{B})$. The Riemannian barycenter has shown strong theoretical and practical interest in Riemannian data analysis [41], which justifies its usage in this context. By definition, $\mathfrak{G}$ is the point on the manifold that minimizes inertia in terms of the Riemannian metric defined in equation 2. The definition is trivially extensible to a weighted Riemannian barycenter, noted $Bar_{\boldsymbol{w}}(\{P_i\}_{i \leq N})$ or $Bar_{\boldsymbol{w}}(\mathcal{B})$, where the weights $\boldsymbol{w} := \{w_i\}_{i \leq N}$ respect the convexity constraint:

$$\mathfrak{G} = Bar_{\boldsymbol{w}}(\{P_i\}_{i \leq N}) := arg \min_{G \in \mathcal{S}_*^+} \sum_{i=1}^N w_i \delta_{\mathfrak{R}}^2(G, P_i), \text{ with } \begin{cases} w_i \geq 0 \\ \sum_{i \leq N} w_i = 1 \end{cases} \tag{4}$$

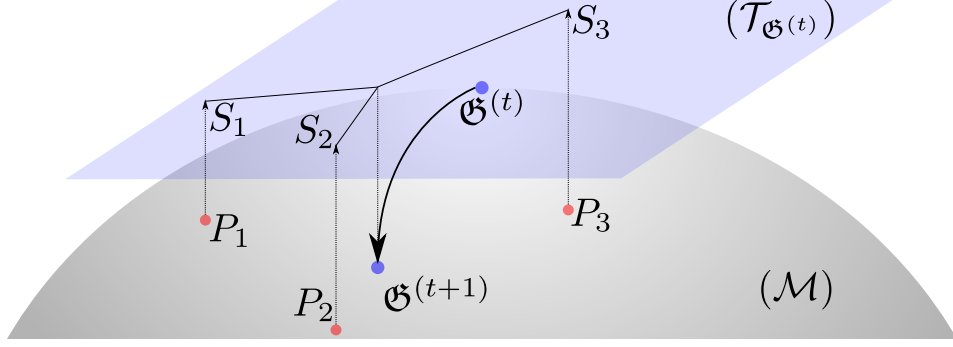

Figure 1: Illustration of one iteration of the Karcher flow [34].

When $N = 2$, i.e. when $\boldsymbol{w} = \{w, 1 - w\}$, a closed-form solution exists, which exactly corresponds to the geodesic between two points $P_1$ and $P_2$, parameterized by $w \in [0, 1]$ [12]:

$$Bar_{(w, 1-w)}(P_1, P_2) \; = \; P_2^{\frac{1}{2}} \left( P_2^{-\frac{1}{2}} P_1 P_2^{-\frac{1}{2}} \right)^{w} P_2^{\frac{1}{2}} \text{, with } w \geq 0 \tag{5}$$

Unfortunately, when $N > 2$, the solution to the minimization problem is not known in closed-form: thus $\mathfrak{G}$ is usually computed using the so-called Karcher flow algorithm [34, 49] , which we illustrate in Figure 1 . In short, the Karcher flow is an iterative process in which data points projected using the logarithmic mapping (equation 3b) are averaged in tangent space and mapped back to the manifold using the exponential mappings (equation 3a) , with a guaranteed convergence on a manifold with constant negative curvature, which is the case for $\mathcal{S}_*^+$ [34]. The initialization of $\mathfrak{G}$ is arbitrary, but a reasonable choice is the arithmetic mean. A key point is that convergence is guaranteed on a manifold with constant negative curvature, which is the case for the SPD manifold $\mathcal{S}_*^+$ [34]. Another point of interest is that selecting $K = 1$ (that is, only one iteration of the flow) and $\alpha = 1$ (unit step size) in the Karcher algorithm , corresponds exactly to the barycenter from the Log-Euclidean metric viewpoint [41]. We actually use this setting in the layer: as the batch barycenter is but a noisy estimation of the true barycenter, a lax approximation is sufficient, and also allows for much faster inference.

## 2.3 Centering SPD matrices using parallel transport

The Euclidean batchnorm involves centering and biasing the batch $\mathcal{B}$, which is done via subtraction and addition. However on a curved manifold, there is no such group structure in general, so these seemingly basic operations are ill-defined. To shift SPD matrices around their mean $\mathfrak{G}$ or towards a bias parameter $G$, we propose to rather use parallel transport on the manifold [2]. In short, the parallel transport (PT) operator $\Gamma_{P_1 \to P_2}(S)$ of a vector $S \in \mathcal{T}_{P_1}$ in the tangent plane at $P_1$, between $P_1, P_2 \in \mathcal{S}_*^+$ defines the path from $P_1$ to $P_2$ such that $S$ remains parallel to itself in the tangent planes along the path. The geodesic $\gamma_{P_1 \to P_2}$ is itself a special case of the PT, when $S$ is chosen to be the direction vector $\gamma'_{P_1 \to P_2}(0)$ from $P_1$ to $P_2$. The expression for PT is known on $\mathcal{S}_*^+$:

$$\forall S \in \mathcal{T}_{P_1}, \; \Gamma_{P_1 \to P_2}(S) = (P_2 P_1^{-1})^{\frac{1}{2}} \, S \, (P_2 P_1^{-1})^{\frac{1}{2}} \in \mathcal{T}_{P_2} \tag{6}$$

The equation above defines PT for tangent vectors, while we wish to transport points on the manifold. To do so, we simply project the data points to the tangent space using the logarithmic mapping , parallel transport the resulting vector from Eq. 6 which we then map back to the manifold using exponential mapping . It can be shown (see [47], appendix C for a full proof) that the resulting operation, which we call SPD transport, turns out to be exactly the same as the formula above, which is not an obvious result in itself. By abuse of notation, we also use $\Gamma_{P_1 \to P_2}$ to denote the SPD transport. Therefore, we can now define the centering of a batch of matrices $\{P_i\}_{i \leq N}$ with Riemannian barycenter $\mathfrak{G}$ as the PT from $\mathfrak{G}$ to the identity $I_d$, and the biasing of the batch towards a parametric SPD matrix $G$ as the PT from $I_d$ to $G$.

**Batch centering and biasing** We now have the tools to define the batch centering and biasing:

$$\text{Centering from } \mathfrak{G} := Bar(\mathcal{B}): \forall i \leq N, \ \bar{P}_i = \Gamma_{\mathfrak{G} \to I_d}(P_i) = \mathfrak{G}^{-\frac{1}{2}} \ P_i \ \mathfrak{G}^{-\frac{1}{2}} \tag{7a}$$

$$\text{Biasing towards parameter } G: \ \forall i \leq N, \ \tilde{P}_i = \Gamma_{I_d \to G}(\bar{P}_i) = G^{\frac{1}{2}} \ \bar{P}_i \ G^{\frac{1}{2}} \tag{7b}$$

## 3  Batchnorm for SPD data

In this section we introduce the Riemannian batch normalization (Riemannian BN, or RBN) algorithm for SPD matrices. We first briefly recall the basic architecture of an SPD neural network.

### 3.1  Basic layers for SPD neural network

The SPDNet architecture mimics that of classical neural networks with a first stage devoted to compute a pertinent representation of the input data points and a second stage which allows to perform the final classification. The particular structure of $\mathcal{S}_*^+$, the manifold of SPD matrices, is taken into account by layers crafted to respect and exploit this geometry. The layers introduced in [28] are threefold:

The BiMap (bilinear transformation) layer, analogous to the usual dense layer; the induced dimension reduction eases the computational burden often found in learning algorithms on SPD data:

$$X^{(l)} = W^{(l)^T} P^{(l-1)} W^{(l)} \text{ with } W^{(l)} \text{ semi-orthogonal} \tag{8}$$

The ReEig (rectified eigenvalues activation) layer, analogous to the ReLU activation; it can also be seen as a eigen-regularization, protecting the matrices from degeneracy:

$$X^{(l)} = U^{(l)} \max(\Sigma^{(l)}, \epsilon I_n) U^{(l)^T} \text{ , with } P^{(l)} = U^{(l)} \Sigma^{(l)} U^{(l)^T} \tag{9}$$

The LogEig (log eigenvalues Euclidean projection) layer:

$$X^{(l)} = vec(\ U^{(l)} \log(\Sigma^{(l)}) U^{(l)^T}\ ) \text{ , with again } U^{(l)} \text{ the eigenspace of } P^{(l)} \tag{10}$$

This final layer has no Eucidean counterpart: its purpose is the projection and vectorization of the output feature manifold to a Euclidean space, which allows for further classification with a traditional dense layer. As stated previously, it is possible to envision different formulations for each of the layers defined above (see [23, 52, 37] for varied examples). Our following definition of the batchnorm can fit any formulation as it remains an independent layer.

### 3.2  Statistical distribution on SPD matrices

In traditional neural nets, batch normalization is defined as the centering and standardization of the data within one batch, followed by the multiplication and addition by parameterized variance and bias, to emulate the data sampling from a learnt Gaussian distribution. In order to generalize to batches of SPD matrices, we must first define the notion of Gausian density on $\mathcal{S}_*^+$. Although this definition has not yet been settled for good, several approaches have been proposed. In [33], the authors proceed by introducing mean and variance as second- and fourth-order tensors. On the other hand, [43] derive a scalar variance. In another line of work synthesized in [9], which we adopt in this work, the Gaussian density is derived from the definition of maximum entropy on exponential families using information geometry on the cone of SPD matrices. In this setting, the natural parameter of the resulting exponential family is simply the Riemannian mean; in other words, this means the notion of variance, which appears in the Eucidean setting, takes no part in this definition of a Gaussian density on $\mathcal{S}_*^+$. Specifically, such a density $p$ on SPD matrices $P$ of dimension $n$ writes:

$$p(P) \propto det(\alpha \ \mathfrak{G}^{-1}) e^{-tr(\alpha \ \mathfrak{G}^{-1} P)} \text{ , with } \alpha = \frac{n+1}{2} \tag{11}$$

In the equation above, $\mathfrak{G}$ is the Riemannian mean of the distribution. Again, there is no notion of variance: the main consequence is that a Riemannian BN on SPD matrices will only involve centering and biasing of the batch.

## 3.3 Final batchnorm algorithm

While the normalization is done on the current batch during training time, the statistics used in inference are computed as running estimations. For instance, the running mean over the training set, noted $\mathfrak{G}_{\mathcal{S}}$, is iteratively updated at each batch. In a Euclidean setting, this would amount to a weighted average between the batch mean and the current running mean, the weight being a momentum typically set to $0.9$. The same concept holds for SPD matrices, but the running mean should be a Riemannian mean weighted by $\eta$, i.e. $Bar_{(\eta, 1-\eta)}(\mathfrak{G}_{\mathcal{S}}, \mathfrak{G}_{\mathcal{B}})$, which amounts to transporting the running mean towards the current batch mean by an amount $(1 - \eta)$ along the geodesic. We can now write the full RBN algorithm 1. In practice, Riemannian BN is appended after each BiMap layer in the network.

---

**Algorithm 1** Riemannian batch normalization on $\mathcal{S}_*^+$, training and testing phase

---

**TRAINING PHASE**
**Require: batch of $N$ SPD matrices $\{P_i\}_{i \leq N}$, running mean $\mathfrak{G}_{\mathcal{S}}$, bias $G$, momentum $\eta$**
1: $\mathfrak{G}_{\mathcal{B}} \leftarrow Bar(\{P_i\}_{i \leq N})$                         // compute batch mean
2: $\mathfrak{G}_{\mathcal{S}} \leftarrow Bar_{\eta}(\mathfrak{G}_{\mathcal{S}}, \mathfrak{G}_{\mathcal{B}})$                 // update running mean
3: **for** $i \leq N$ **do**
4:     $\bar{P}_i \leftarrow \Gamma_{\mathfrak{G}_{\mathcal{B}} \rightarrow I_d}(P_i)$                  // center batch
5:     $\tilde{P}_i \leftarrow \Gamma_{I_d \rightarrow G}(\bar{P}_i)$                   // bias batch
6: **end for**
    **return normalized batch** $\{\tilde{P}_i\}_{i \leq N}$

**TESTING PHASE**
**Require: batch of $N$ SPD matrices $\{P_i\}_{i \leq N}$, final running mean $\mathfrak{G}_{\mathcal{S}}$, learnt bias $G$**
1: **for** $i \leq N$ **do**
2:     $\bar{P}_i \leftarrow \Gamma_{\mathfrak{G}_{\mathcal{S}} \rightarrow I_d}(P_i)$                  // center batch using set statistics
3:     $\tilde{P}_i \leftarrow \Gamma_{I_d \rightarrow G}(\bar{P}_i)$                   // bias batch using learnt parameter
4: **end for**
    **return normalized batch** $\{\tilde{P}_i\}_{i \leq N}$

---

# 4 Learning the batchnorm

The specificities of a the proposed batchnorm algorithm are the non-linear manipulation of manifold values in both inputs and parameters and the use of a Riemannian barycenter. Here we present the two results necessary to correctly fit the learning of the RBN in a standard back-propagation framework.

## 4.1 Learning with SPD constraint

The bias parameter matrix $G$ of the RBN is by construction constrained to the SPD manifold. However, noting $\mathcal{L}$ the network's loss function, the usual Euclidean gradient $\frac{\partial \mathcal{L}}{\partial G}$, which we note $\partial G_{eucl}$, has no particular reason to respect this constraint. To enforce it, $\partial G_{eucl}$ is projected to the tangent space of the manifold at $G$ using the manifold's tangential projection operator $\Pi \mathcal{T}_G$, resulting in the tangential gradient $\partial G_{riem}$. The update is then obtained by computing the geodesic on the SPD manifold emanating from $G$ in the direction $\partial G_{riem}$, using the exponential mapping defined in equation 3a. Both operators are known in $\mathcal{S}_*^+$ [50]:

$$\forall P, \ \Pi \mathcal{T}_G(P) = G \frac{P + P^T}{2} G \in \mathcal{T}_G \subset \mathcal{S}^+ \tag{12}$$

We illustrate this two-step process in Figure 2, explained in detail in [24], which allows to learn the parameter in a manifold-constrained fashion. However, this is still not enough for the optimization of the layer, as the BN involves not simply $G$ and $\mathfrak{G}$, but $G^{\frac{1}{2}}$ and $\mathfrak{G}^{-\frac{1}{2}}$, which are structured matrix functions of $G$, i.e. which act non-linearly on the matrices' eigenvalues without affecting its associated eigenspace. The next subsection deals with the backpropagation through such functions.

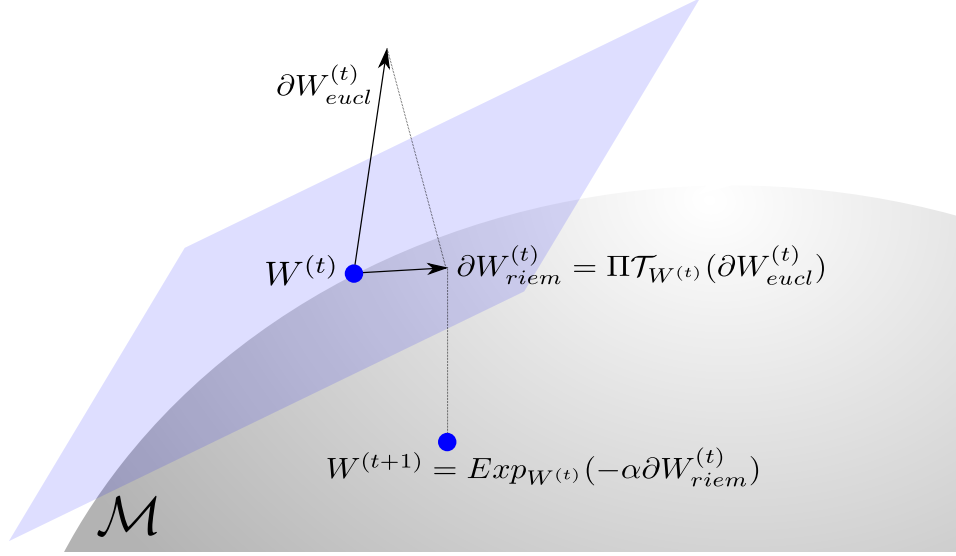

Figure 2: Illustration of manifold-constrained gradient update. The Euclidean gradient is projected to the tangent space, then mapped to the manifold.

## 4.2 Structured matrix backpropagation

Classically, the functions involved in the chain rule are vector functions in $\mathbb{R}^n$ [35], whereas we deal here with structured (symmetric) matrix functions in the $\mathcal{S}_*^+$, specifically the square root $(\cdot)^{\frac{1}{2}}$ for the bias and the inverse square root $(\cdot)^{-\frac{1}{2}}$ for the barycenter (in equations 7b 7a). A generalization of the chain rule to $\mathcal{S}_+^*$ is thus required for the backpropagation through the RBN layer to be correct. Note that a similar requirement applies to the ReEig and LogEig layers, respectively with a threshold and log function. We generically note $f$ a monotonous non-linear function; both $(\cdot)^{\frac{1}{2}}$ and $(\cdot)^{-\frac{1}{2}}$ check out this hypothesis. A general formula for the gradient of $f$, applied on a SPD matrix' eigenvalues $(\sigma_i)_{i \leq n}$ grouped in $\Sigma$'s diagonal, was independently developed by [32] and [13]. In short: given the function $P \longmapsto X := f(P)$ and the succeeding gradient $\frac{\partial L^{(l+1)}}{\partial X}$, the output gradient $\frac{\partial L^{(l)}}{\partial P}$ is:

$$\frac{\partial L^{(l)}}{\partial P} = U \left( L \odot (U^T (\frac{\partial L^{(l+1)}}{\partial X})U) \right) U^T \tag{13}$$

The equation above, also decribed in [39], is called the Daleckiĭ-Kreĭn formula and dates back to 1956, (but was translated from Russian 9 years later), predating the other formulation by 60 years. It involves the eigenspace $U$ of the input matrix $P$, and the Loewner matrix $L$, or finite difference matrix defined by:

$$L_{ij} = \begin{cases} \frac{f(\sigma_i) - f(\sigma_j)}{\sigma_i - \sigma_j} & \text{if } \sigma_i \neq \sigma_j \\ f'(\sigma_i) & \text{otherwise} \end{cases} \tag{14}$$

In the case at hand, $\left( (\cdot)^{-\frac{1}{2}} \right)' = -\frac{1}{2}(\cdot)^{-\frac{3}{2}}$ and $\left( (\cdot)^{\frac{1}{2}} \right)' = \frac{1}{2}(\cdot)^{-\frac{1}{2}}$. We credit [25] for first showing the equivalence between the two cited formulations, of which we expose the most concise.

In summary, the Riemannian barycenter (approximation via the Karcher flow for a batch of matrices, or exact formulation for two matrices), the parallel transport and its extension on the SPD manifold, the SPD-constrained gradient descent and the derivation of a non-linear SPD-valued structured function's gradient allow for training and inference of the proposed Riemannian batchnorm algorithm.

Table 1: Accuracy comparison of SPDNet, SPDNetBN and FCNs on NATO radar data, in function of amount of training data.

| Model | SPDNet | SPDNetBN | FCN | | MRDRM |
|---|---|---|---|---|---|
| # Parameters | $\sim 500$ | $\sim 500$ | $\sim 10000$ | $\sim 500$ | - |
| Acc. (all data) | $72.6\% \pm 0.61$ | $82.3\% \pm 0.80$ | $\mathbf{88.7}\% \pm 0.83$ | $73.4\% \pm 3.66$ | $69.7\% \pm 1.12$ |
| Acc. (10% data) | $69.1\% \pm 0.97$ | $\mathbf{77.7}\% \pm 0.95$ | $65.6\% \pm 2.74$ | $61.1\% \pm 3.50$ | $67.1\% \pm 2.17$ |

# 5 Experiments

Here we evaluate the gain in performance of the RBN against the baseline SPDNet on different tasks: radar data classification, emotion recognition from video, and action recognition from motion capture data. We call the depth $L$ of an SPDNet the number of BiMap layers in the network, and denote the dimensions as $\{n_0, \cdots, n_L\}$. The vectorized input to the final classification layer is thus of length $n_L^2$. All networks are trained for 200 epochs using SGD with momentum set to 0.9 with a batch size of 30 and learning rate $5e^{-3}$, $1e^{-2}$ or $5e^{-2}$. We provide the data in a pre-processed form alongside the PyTorch [40] code for reproducibility purposes. We call SPDNetBN an SPDNet using RBN after each BiMap layer. Finally, we also report performances of shallow learning method on SPD data, namely a minimum Riemannian distance to Riemannian mean scheme (MRDRM), described in [7], in order to bring elements of comparison between shallow and deep learning on SPD data.

## 5.1 Drones recognition

Our first experimental target focuses on drone micro-Doppler [21] radar classification. First we validate the usage of our proposed method over a baseline SPDNet, and also compare to state-of-the-art deep learning methods. Then, we study the models' robustness to lack of data, a challenge which, as stated previously, plagues the task of radar classification and also a lot of different tasks. Experiments are conducted on a confidential dataset of real recordings issued from the NATO organization [1] . To spur reproducibility, we also experiment on synthetic, publicly available data.

**Radar data description**  A radar signal is the result of an emitted wave reflected on a target; as such, one data point is a time-series of $N$ values, which can be considered as multiple realizations of a locally stationary centered Gaussian process, as done in [20]. The signal is split in windows of length $n = 20$, the series of which a single covariance matrix of size $20 * 20$ is sampled from, which represents one radar data point. The NATO data features 10 classes of drones, whereas the synthetic data is generated by a realistic simulator of 3 different classes of drones following the protocol described in [14]. We chose here to mimick the real dataset's configuration, i.e. we consider a couple of minutes of continuous recordings per class, which correspond to 500 data points per class.

**Comparison of SPDNetBN against SPDNet and radar state-of-the-art**  We test the two SPD-based models in a $\{20, 16, 8\}$, 2-layer configuration for the synthetic data, and in a $\{20, 16, 14, 12, 10, 8\}$, 5-layer configuration for the NATO data, over a 5-fold cross-validation, split in a train-test of $75\% - 25\%$. We also wish to compare the Riemannian models to the common Euclidean ones, which currently consitute the state-of-the-art in micro-Doppler classification. We compare two fully convolutional networks (FCN): the first one is used as given in [14]; for the second one, the number of parameters is set to approximately the same number as for the SPD neural networks, which amounts to an unusually small deep net. All in all, the SPDNet, SPDNetBN and small FCN on the one hand, and the full-size FCN on the other hand respectively have approximately 500 and 10000 parameters. Table 1 reports the average accuracies and variances on the NATO data. We observe a strong gain in performance on the SPDNetBN over the SPDNet and over the small FCN, which validates the usage of the batchnorm along with the exploitation of the geometric structure underlying the data. All in all, we reach better performance with much fewer parameters.

Finally, in the interest of convergence analysis, we also report learning curves for the model's accuracy with and without Riemannian batchnorm in figure 3.

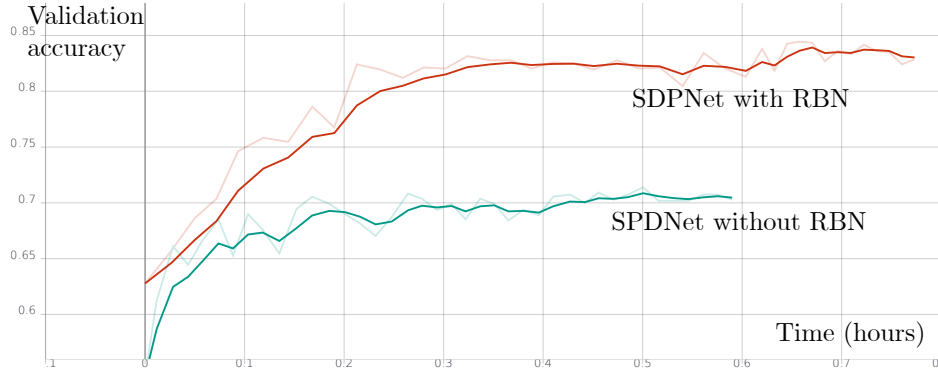

Figure 3: Test accuracy on the NATO dataset of the SPDNet with and without RBN, measured in hours. The RBN exhibits a steeper learning curve. For the same number of epochs, it does take more time overall, but reaches better accuracy much faster, allowing to reduce the number of epochs.

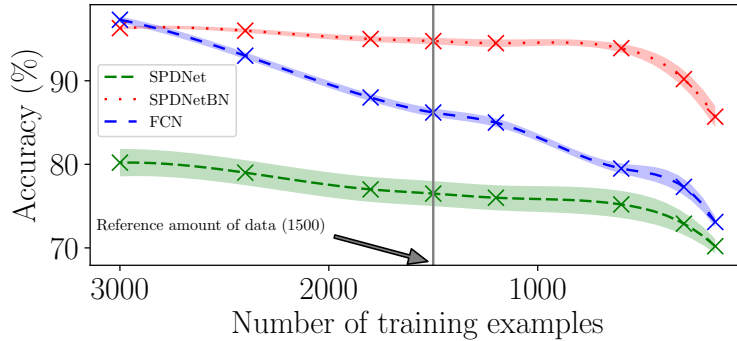

Figure 4: Performance of all models in function of the amount of synthetic radar data. The SPDNetBN model outperforms the other ones and continues to work even with a little fraction of the train data.

**Robustness to lack of data**     As stated previously, it is of great interest to consider the robustness of learning algorithms when faced with a critically low amount of data. The last line in table 1 shows that when given only $10\%$ of available training data, the SPD-based models remain highly robust to the lack of data while the FCNs plummet. Further, we study robustness on synthetic data, artificially varying the amount of training data while comparing performance over the same test set. As the simulator is unbounded on potential training data, we also increase the initial training set up to double its original size. Results are reported in Figure 4. We can conclude from these that the SPDNetBN both exhibits higher robustness to lack of data and performs much better than the state-of-the-art deep method with much fewer parameters. When the available training data allowed skyrockets, we do observe that the FCN comes back to par with the SPDNetBN to the point of outperforming it by a small margin in the extremal scenario; in the meantime, the SPDNet lags behind by a large margin to the SPDNetBN, which thus seems to benefit strongly from the normalization. In any case, the manifold framework seems well suited in a scarce data learning context, especially considering the introduced normalization layers, which again pinpoints the interest of taking into account the geometric structure of the data, all the while without introducing prior knowledge during training.

## 5.2   Other experiments

Here we validate the use of the RBN on a broader set of tasks. We first clarify we do not necessarily seek state-of-the-art in the general sense for the following tasks, but rather in the specific case of the family of SPD-based methods. Our own implementation (as an open PyTorch library) of the SPDNet's performances match that in [28], ensuring a fair comparison.

Table 2: Accuracy comparison of SPDNet with and without Riemannian BN on the AFEW dataset.

| Model architecture | $\{400, 50\}$ | $\{400, 100, 50\}$ | $\{400, 200, 100, 50\}$ | $\{400, 300, 200, 100, 50\}$ |
|---|---|---|---|---|
| SPDNet | 29.9% | 31.2% | 34.5% | 33.7% |
| SPDNetBN (ours) | **34.9**% | **35.2**% | **36.2**% | **37.1**% |

Table 3: Accuracy comparison of SPDNet with and without Riemannian BN on the HDM05 dataset.

| Model architecture | SPDNet | SPDNetBN (ours) |
|---|---|---|
| $\{93, 30\}$ | 61.6%±1.35 | **65.2**% ± 1.15 |

**Emotion recognition** In this section we experiment on the AFEW dataset [22], which consists of videos depicting 7 classes of emotions; we follow the setup and protocol in [28]. Results for 4 architectures are summarized in table 2. In comparison, the MRDRM yields a $20.5\%$ accuracy. We observe a consistent improvement using our normalization scheme. This dataset being our largest-scale experiment, we also report the increase in computation time using the RBN, specifically for the deepest net: one training lasted on average $81s$ for SPDNet, and $88s$ $(+8.6\%)$ for SPDNetBN.

**Action recognition** In this section we experiment on the HDM05 motion capture dataset. We use the same experimental setup as in [28] results are shown in table 3. Note that all tested models exhibit noticeable variance depending on the weights initialization and the initial random split of the dataset; the results displayed were obtained by setting a fixed seed of 0 for both. In comparison, the MRDRM yields a $27.3\% \pm 1.06$ accuracy. Again, we validate a better performance using the batchnorm.

## Conclusion

We proposed a batch normalization algorithm for SPD neural networks, mimicking the orginal batchnorm in Euclidean neural networks. The algorithm makes use of the SPD Riemannian manifold's geometric structure, namely the Riemannian barycenter, parallel transport, and manifold-constrained backpropagation through non-linear structured functions on SPD matrices. We demonstrate a systematic, and in some cases considerable, performance increase across a diverse range of data types. An additional observation is the better robustness to lack of data compared to the baseline SPD neural network and to a state-of-the-art convolutional network, as well as better performance than a well-used, more traditional Riemannian learning method (the closest-barycenter scheme). The overall performances of our proposed SPDNetBN makes it a suitable candidate in learning scenarios where data is structured, scarce, and where model size is a relevant issue.

## Footnotes

[1] We would like to thank the NATO working group SET245 for providing the drone micro-Doppler database and allowing for publication of classification results.

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
