[Supplementary Material]

# Supplementary material

**Daniel Brooks**

## 1 Reproducibility

The code reproducing the experiments can be found in the archive *RiemBN_pytorch*, which is organized following the PyTorch module and functional logic:

1. *spd/*: building blocks for the SPDNetBN architecture and training:
   *functional.py*: core machinery for inference and backpropagation in a SPDNetBN;
   *nn.py*: modules used to build a SPDNetBN architecture;

2. *cplx/*: building blocks for the complex data handling (only useful for radar data);
   *functional.py*: core machinery for inference and backpropagation in a complex neural net [4];
   *nn.py*: modules used to build a complex net architecture;

3. *experiments/*:
   *radar.py*: code for lauching experiments on the synthetic radar dataset (results on the real dataset are not reproducible beacause of data confidentiality issues);
   *hdm05.py*: code for lauching experiments on the HDM05 dataset;
   *afew.py*: code for lauching experiments on the AFEW dataset;
   *radar_mrdrm.py*: code for lauching minimum Riemannian distance to Riemannian mean (MRDRM) experiment on the synthetic radar dataset (results on the real dataset are not reproducible beacause of data confidentiality issues);
   *hdm05_mrdrm.py*: code for lauching MRDRM experiment on the HDM05 dataset;
   *afew_mrdrm.py*: code for lauching MRDRM experiment on the AFEW dataset;
   *data/*: the folder to copy the data to; the others are made available for download [1].
   Radar, HDM05 and AFEW datasets respectively weigh 47Mb, 139Mb and 1.3Gb.

To launch the code in the *experiments/* directory, execute for instance *python radar.py*. The code was developed in Python3 and tested under PyTorch [3] v0.4.1 on CUDA version 8.0.61 and run on a laptop i7-6700HQ CPU.

## 2 Details on the Karcher flow

Here we give a brief description of the Karcher flow algorithm 1 described in the paper, as well as an illustration of one step of the algorithm 1. the so-called Karcher flow algorithm [2, 5]

## 3 Description of the experiments

All models in the experiments are trained for 200 epochs on batches of size 30 using vanilla SGD for HDM05 and AFEW experiments, and with momentum set to 0.9 and weight decay $5e - 4$ for radar experiments, and a common learning rate of $5e^{-2}$. All provided data except for radar are already processed to SPD matrices (pre-processing modules are included in the package).

**Algorithm 1** Karcher flow [2] to compute the Riemannian mean of $N$ SPD matrices

**Require:** data points $\{P_i\}_{i \leq N}$, iterations $K$, step $\alpha$
1: $\mathfrak{G} \leftarrow \sum_{i \leq N} P_i$
2: **for** $k \leq K$ **do**
3: $\quad G \leftarrow \frac{1}{N} \sum_{i \leq N} Log_{\mathfrak{G}}(P_i)$
4: $\quad \mathfrak{G} \leftarrow Exp_{\mathfrak{G}}(\alpha G)$
5: **end for**
6: **return** $\mathfrak{G}$

Figure 1: Illustration of one iteration of the Karcher flow [2]. Data points $P_i$ are logarithmically mapped to the $S_i$ on the tangent space at $\mathfrak{G}^{(t)}$. The $S_i$ are then arithmetically averaged, the result of which is exponentially mapped back to the manifold, yielding $\mathfrak{G}^{(t+1)}$.

**Radar** The amount of data provided in the archive corresponds to the maximal amount mentioned in the paper, which results to 3000 SPD data points of size $20 \times 20$ equally distributed in the 3 classes.

**AFEW** The AFEW dataset consists of 2118 SPD matrices of size $400 * 400$ split in $1747 + 371$ fixed training and validation sets, each matrix in the dataset summarizing the pixel cross-correlations of a video clip scaled down to frames of size $20 * 20$.

**HDM05** The HDM05 dataset consists in motion capture data (MoCap) depicting various actions performed by 5 human actors. The actions are divided in 14 main action classes, which are subdivided in 130 classes. We perform the experiments in the 130-class setting. One data point is a sequence of $m$ body states evolving through time, one body state being a frame of $3D$ coordinates of the $n_{joints} = 31$ joints constituting the body. For each body state, we consider the $n = 3 * n_{joints} = 93$-dimensional feature vector concatenating all coordinates; the whole sequence is then described by its centered $n * n$ joint covariance, ie the summed centered covariances of each frame feature vector. The resulting matrix can be SPD iff $n \leq m$. The database contains 2337 sequences. Some of the 130 classes being vastly under-represented, we select those with at least 5 sequences yielding SPD descriptors (as done in [1]), which trims the dataset down to 2086 points scattered throughout 117 classes.

## 4 Implementation details

Here we list some details of interest to reproduce the results:

1. Training did not benefit from GPU acceleration, seemingly bottlenecked at eigenvalue operations: computation time desribed in the paper is not decreased using an Nvidia GTX 1070M;

2. Symmetric matrix vectorization: following the Euclidean mapping, it is possible to vectorize only the upper triangular part of the matrix (and normalize the outer-diagonal coefficients by $\sqrt{2}$ to conserve the norm). We conducted all experiments in both settings and

observed no significant impact, except for the parametric centering, which seemed to suffer from a smaller representation dimension. The reported results use full matrix vectorization;

3. Matrix orthonormalization: for reasons we are unsure of, performance benefitted from a home-made implementation of the classical Gram-Schmidt (CGS) process compared to the built-in QR decomposition, especially in the AFEW experiments; again, the exception was the parametric centering which followed the opposite rule; however for fairness of evaluation, we reported the results for the same configuration, i.e. our own CGS;

4. Floating-point precision: as in previous works using structured matrix differentiation, working in double precision was paramount for smooth convergence in all cases; in practice, we observe the precision is most important in the computation of the Loewner matrix, i.e. in the backprop of structured matrix functions, and specifically while inverting differences in eigenvalues;

5. Since we deal with SPD matrices, eigen-decomposition (EIG) is equivalent to singular value decomposition (SVD): the latter being more stable and cheaper to compute, it is possible to use an $SVD$ algorithm, as we and previous works on SPD matrices do. That being said, the matrix exponentiation during the computation of the Riemannian barycenter needs switching to an $EIG$ algorithm as negative eigenvalues can exist in the symmetric matrices.

## Footnotes

[1]The synthetic radar, HDM05 and AFEW datasets may be found at `https://www.dropbox.com/s/dfnlx2bnyh3kjwy/data.zip?dl=0`