[Reviews · NeurIPS 2019]

Reviewer 1



POST-REBUTTAL: authors have answered my questions satisfactorily, hence I am increasing my score to 7. I enjoyed reading this clearly written piece. It's a little bit of a misnomer to call this method "batch normalization", since it does not normalize the "spread", but only the "center". This is justified by the authors describing distributions over SPD matrices (sec 3.2). (Side question: what's the relationship between eq (10) and Wishart distribution? Are they the same, but one with the Riemannian metric? Wishart is also a maximum entropy distribution over SPD.) However, maximum entropy distribution depends not only on the space, but also which sufficient statistics one choses. I do not see an inherent reason why variance-like quantity should be ignored. I suggest changing the name of the method to "batch centering" to avoid confusion. Since traditional BN improves on learning rate, I wish to see learning curves with this method. How does this method compare (final performance, and convergence speed) to a naive BN the projection to closest point on the manifold? (perhaps this is a stupid question, but please explain) Also, the computational complexity of this method for forward and backward pass are not obvious. Can you include them?

Reviewer 2



Originality: this paper contains an original contribution, which is tightly based on [22]'s SPD networks, where the hidden representations are symmetric positive definite (SPD) matrices. The originality lies in that the authors adapted batch normalization which is usually performed on real vectors to the SPD representations. Notice that in their batch normalization, the variance is not normalized. The authors may consider to use a different term (e.g. batch centering) because of this. Quality: the proposed method is particularly useful for SPD networks, and could be useful in other networks with SPD hidden representations. The author should mention the complexity explicitly in computing the exponential map and its inverse. How much computational overhead is involved in applying your batch normalization? Clarity: the writing is satisfactory and the algorithms are clearly presented. But the math formulae can be greatly improved. In particular in the beginning there should be a small section to explain the Riemannian geometry of SPD matrices. Significance: this is a deep learning paper based on Riemannian geometry. It can be useful in the particular deep learning architecture SPD networks [22]. It may be also interesting from an application perspective (applying SPDNet to EEG/MRI datasets). From the mathematical standpoint the novelty is limited. Overall I feel that the potential significance is average because of these limitations. minor comments: The authors can cite related references is matrix Bregman divergence, Bures distance in deep learning, as well as log-det divergence. Notice that for Bregman divergence one can have the notion of variance. eq.(1) explain what is "log" (use \log instead of log) Is that matrix logarithm? L39 It is not clear what "each layer processes a point on the SPD manifold" means The paper seems to be finished in a rush, table 1 & table 3 are formatted in a bad way.

Reviewer 3



The paper is well written and very comprehensible. Necessary basics are explained and the experimental results give evidence that the proposed method increases classification performance. The experiments seem to be done very thoroughly. While the results on the NATO radar dataset is not very outstanding, the method seems to outperform the baseline SPDNet on the AFW and the HDM05 dataset. Overall, good work with some minor errors: - line 24: symmetric positive definite (no capitalization) - throughout the paper: mathoperators (like log, exp, argmin, ...) can be defined using \DeclareMathOperator{\argmin}{argmin} - algorithm 1: batch and normalized (no capitalization)

[Author Response · NeurIPS 2019]

As R#1 and R#2 suggested, we agree to rename the method to "Riemannian batch centering" (RBC), or "Riemannian
variance-free batchnorm".

**Q :** *Also, the computational complexity of this method for forward and backward pass are not obvious. Can you include*
*them?* (R#1) / *Ths authors should [...] make the computational overhead very clear.* (R#2)

→ In the paper we report overhead in the SPDNet from using our proposed RBC for the deepest net on the most
computationally time-demanding experiment, the AFEW dataset (lines 277-279): the overhead for one epoch is of
$+8.6\%$ relative time increase, which seems acceptable (all other experiments show a comparable, mostly smaller
overhead). As for the complexity: 1) Riemannian barycenter: we use only one step in the Karcher flow, which involves
one log mapping and one exp mapping: the intuition is that since the batch barycenter is but a noisy estimation of the
true barycenter, a lax approximation is sufficient, and also allows for much faster inference; in practice, it even works
better than going through multiple steps, and than using zero steps, i.e. the Euclidean mean; 2) SPD transport: the bulk
of the complexity comes from computing the inverse square root and square root of the Riemannian barycenter and the
learnt bias, respectively. Since they are SPD, the complexity, both in inference and backprop (backprop is only required
for the bias matrix) is equivalent to that of the already existing ReEig and LogEig layers (i.e., applying a non-linear
function to the diagonal matrix of singular values obtained through SVD); 3) Thus, as in the regular SPDNet, the
complexity mostly resides in the SVD of batch of matrices: to reduce this burden, namely in the backprop, SVD results
are stored during the training when they are to be re-used (using the Pytorch *save_for_backward* function). In summary,
the RBC requires the SVD of the batch, plus two additional SVDs, one for the barycenter and one for the bias; all other
operations are scalar operations and matrix multiplications. Complexity will be further discussed in a final version.

**Q :** *Side question: what's the relationship between eq (10) and Wishart distribution?* (R#1)

→ Although the similar formulae hint to some link, they are obtained differently and thus don't represent the same
distributions: Wishart deals with data dispersion matrices $XX^T$, whereas the proposed definition stems from defining
the entropy as the Legendre transform of the free energy, which itself is the negative log of the cone's characteristic
function, i.e. the Laplace transform between dual coordinates. See the referenced literature, along with [J. Faraut.
Analysis on symmetric cones] and [F. Barbaresco. Jean-Louis Koszul and the Elementary Structures of Information
Geometry].

**Q :** *Report on learning curve* (R#1)

→ See figure 1: the RBC does seem to provide a steeper learning curve. For the same number of epochs, we see the
RBC takes more time overall, but reaches better accuracy much faster, allowing to reduce the number of epochs. Note
that tests were re-run with a deeper net on a more challenging configuration than that in the original submission, where
SPDNet with RBC remains stable but drastically drops without.

**Q :** *Compare with traditional BN with projections to the*
*manifold* (R#1)

→ The comparison is definitely of interest: the closest
point on the tangent bundle to each SPD matrix is its
matrix log, and it is indeed possible to see matrices as
standard $2D$ images and use a standard batchnorm. This
experiment on the NATO radar data yields a score of
$74.3\% \pm 2.01$, compared to the $87.2\% \pm 1.06$ reported in
the paper. Furthermore, not using the matrix log yields
even worse and less stable performance ($58.6\% \pm 2.17$).
We believe both results further justify the use of Rie-
mannian geometry when handling SPD data (as expected
given the literature).

Figure 1: Validation accuracy on the NATO radar dataset in function of physical time. The two curves show the same number of epochs.

**Q :** *eq.(1) explain what is "log"* (R#2)

→ It is the matrix log, as recalled a few lines below (line 84); we can move the definition a few lines up.

**Q :** *L39 It is not clear what "each layer processes a point on the SPD manifold" means* (R#2)

→ We meant to contrast with "traditional" networks, which deal with points in an Euclidean space.

We thank R#3 for spotting the typos, and will make sure to fix them, and all reviewers for other general commentaries
such as the addition of background explanations on Riemannian geometry and the mentioning of alternate work based
on different metrics ([K. Sun, P. Koniusz, Z. Wang, Fisher-Bures Adversary Graph Convolutional Networks], [A.
Siahkamari, V. Saligrama, D. Castanon, and B. Kulis. Learning Bregman Divergences], log-det divergence...) .

[Meta-Review · NeurIPS 2019]

A novel batch normalization layer adapted to SPD networks, with a clear mathematical derivation and experimental validation. Additional results regarding convergence vs. learning rate would be interesting.